# Differentiable Pareto-Smoothed Weighting
# for High-Dimensional Heterogeneous Treatment Effect Estimation

**Yoichi Chikahara**[1]                    **Kansei Ushiyama**[2*]

[1]NTT Communication Science Laboratories, Kyoto, Japan
[2]The University of Tokyo, Tokyo, Japan

## Abstract

There is a growing interest in estimating heterogeneous treatment effects across individuals using their high-dimensional feature attributes. Achieving high performance in such high-dimensional heterogeneous treatment effect estimation is challenging because in this setup, it is usual that some features induce sample selection bias while others do not but are predictive of potential outcomes. To avoid losing such predictive feature information, existing methods learn separate feature representations using inverse probability weighting (IPW). However, due to their numerically unstable IPW weights, these methods suffer from estimation bias under a finite sample setup. To develop a numerically robust estimator by weighted representation learning, we propose a differentiable Pareto-smoothed weighting framework that replaces extreme weight values in an end-to-end fashion. Our experimental results show that by effectively correcting the weight values, our proposed method outperforms the existing ones, including traditional weighting schemes. Our code is available at https://github.com/ychika/DPSW.

## 1 INTRODUCTION

In this paper, we tackle the problem of estimating heterogeneous treatment effects across individuals from high-dimensional observational data. This problem, which we call high-dimensional heterogeneous treatment effect estimation, offers the following crucial applications: the evaluation of medical treatment effects from numerous attributes [Shalit, 2020] and the assessment of the advertising effects from each user's many attributes [Bottou et al., 2013].

One fundamental difficulty in high-dimensional heterogeneous treatment effect estimation is the sample selection bias induced by *confounders*, i.e., the features of an individual that affect their treatment choices and outcomes. For instance, in the case of medical treatment, age is a possible confounder: Elderly patients avoid choosing surgery due to its risk and generally suffer from higher mortality [Zeng et al., 2022]. Due to their treatment choice imbalance, there are often fewer records of elderly patients who have received surgical treatments. As a result, accurately predicting surgical outcomes is difficult in this age cohort, thus complicating heterogeneous treatment effect estimation.

To address such sample selection bias, it is crucial to determine how to break the dependence of treatment choice on confounders. A promising approach for a high-dimensional setup is representation learning [Shalit et al., 2017], which estimates the potential outcomes under different treatment assignments by extracting the *balanced* feature representation that is learned such that its distribution is identical between treated and untreated individuals.

However, since this approach converts all input features to a single balanced representation, if some features are *adjustment variables* (a.k.a., *risk factors*) [Brookhart et al., 2006], which are unrelated to sample selection bias but useful for outcome prediction, such useful feature information may be inadvertently eliminated, leading to inaccurate treatment effect estimation [Sauer et al., 2013]. This issue is serious, especially in high-dimensional setups where input features often contain adjustment variables. Moreover, due to the lack of prior knowledge about the features, it is difficult for practitioners to correctly separate adjustment variables from confounders. Such feature separation might be impossible if one attempts to input the feature embeddings of a complex object (e.g., texts, images, and graphs) that are constructed from pre-trained generative models, including large language models (LLMs) [Keith et al., 2020].

To fulfill such important but often overlooked needs, several data-driven feature separation methods have been proposed

---

*Work during the summer internship at NTT Communication Science Laboratories.

*Accepted for the 40th Conference on Uncertainty in Artificial Intelligence* (UAI 2024).

[Kuang et al., 2017, Hassanpour and Greiner, 2020, Kuang et al., 2020, Wang et al., 2023]. Among them, *disentangled representations for counterfactual regression* (DRCFR) [Hassanpour and Greiner, 2020] aims to avoid losing useful feature information for heterogeneous treatment effect estimation by learning different representations for confounders and adjustment variables. To achieve this, this method employs a technique of inverse probability weighting (IPW) [Rosenbaum and Rubin, 1983], which performs weighting based on the inverse of the probability called a *propensity score*. However, such weights often take extreme values (especially in high-dimensional setups [Li and Fu, 2017]), and even slight propensity score estimation error may lead to large treatment effect estimation error.

To resolve this issue, we develop a weight correction framework that utilizes a technique in extreme value statistics, called *Pareto smoothing* [Vehtari et al., 2024], which replaces the extreme weight values with the quantiles of generalized Pareto distribution (GPD). Indeed, Zhu et al. [2020] already adopted Pareto smoothing and empirically showed that it can construct a numerically stable estimator of the average treatment effect (ATE) over all individuals.

To estimate heterogeneous treatment effects across individuals, we propose a Pareto-smoothed weighting framework that can be combined with the weighted representation learning approach. However, achieving this goal is difficult because weight correction with Pareto smoothing requires the computation of the rank (position) of each weight value; this computation is non-differentiable and prevents gradient backpropagation. To overcome this difficulty, we utilize a *differentiable ranking* technique [Blondel et al., 2020] and establish a differentiable weight correction framework founded on Pareto smoothing. This idea of combining Pareto smoothing and differentiable ranking, both of which have been studied in completely different fields (i.e., extreme value statistics and differentiable programming), allows for effective end-to-end learning for high-dimensional heterogeneous treatment effect estimation.

**Our contributions** are summarized as follows.

- We propose a differentiable Pareto-smoothed weighting framework that replaces extreme IPW weight values in an end-to-end fashion. To make this weight replacement procedure differentiable, we utilize a *differentiable ranking* technique [Blondel et al., 2020].

- Exploiting differentiability, we build our weight correction framework on the neural-network-based weighted representation learning method (i.e., DR-CFR [Hassanpour and Greiner, 2020]) and perform data-driven feature separation for high-dimensional heterogeneous treatment effect estimation.

- We experimentally show that our method effectively learns the feature representations and outperforms the existing ones, including traditional weighting schemes.

## 2 PRELIMINARIES

### 2.1 PROBLEM SETUP

Suppose we have a sample of $n$ individuals, $\mathcal{D} = \{(a_i, \boldsymbol{x}_i, y_i)\}_{i=1}^n \overset{i.i.d.}{\sim} \mathrm{P}(A, \boldsymbol{X}, Y)$, where $A \in \{0, 1\}$ is a binary treatment ($A = 1$ if an individual is treated; otherwise $A = 0$), $\boldsymbol{X} = [X_1, \ldots, X_d]^\top$ is the $d$-dimensional features (a.k.a., covariates), and $Y$ is an outcome. Let $Y^0$ and $Y^1$ be *potential outcomes* i.e., the outcomes when untreated ($A = 0$) and when treated ($A = 1$), which are given by $Y = AY^1 + (1 - A)Y^0$. A treatment effect for an individual is defined as their differences, i.e., $Y^1 - Y^0$ [Rubin, 1974].

We consider the heterogeneous treatment effect estimation problem, where we take (as input) sample $\mathcal{D}$ and feature values $\boldsymbol{x}$ and output the estimate of a conditional average treatment effect (CATE) conditioned on $\boldsymbol{X} = \boldsymbol{x}$:

$$\mathbb{E}\left[Y^1 - Y^0 \mid \boldsymbol{X} = \boldsymbol{x}\right]. \tag{1}$$

CATE is an average treatment effect in a subgroup of individuals who have identical feature attributes $\boldsymbol{X} = \boldsymbol{x}$.

To estimate the CATE in (1), we make two standard assumptions. One is *conditional ignorability*, $\{Y^0, Y^1\} \perp\!\!\!\perp A \mid \boldsymbol{X}$; this conditional independence is satisfied if features $\boldsymbol{X}$ contain all of the confounders and include only *pretreatment variables*, which are not affected by treatment $A$. The other is *positivity*, $0 < \pi(\boldsymbol{x}) < 1$ for all $\boldsymbol{x}$, where $\pi(\boldsymbol{x}) := \mathrm{P}(A = 1 \mid \boldsymbol{X} = \boldsymbol{x})$ is a conditional distribution model called a propensity score.

Our goal is to achieve a high CATE estimation performance in a high-dimensional setup, where the number of features $d$ is relatively large. Under this setup, removing the sample selection bias by transforming all features $\boldsymbol{X}$ into a single balanced representation might be overly severe, which leads to inaccurate treatment effect estimation. Although several representation learning methods aim to avoid such an overly severe balancing [Kuang et al., 2017, Wang et al., 2023], most are designed to estimate ATE, not CATE. By contrast, the DRCFR method [Hassanpour and Greiner, 2020] focuses on CATE estimation and effectively balances high-dimensional features $\boldsymbol{X}$ by weighted representation learning.

### 2.2 WEIGHTED REPRESENTATION LEARNING

DRCFR [Hassanpour and Greiner, 2020] is based on the graphical model in Figure 1, where treatment $A$ is determined by functions $\Gamma(\boldsymbol{X})$ and $\Delta(\boldsymbol{X})$, while outcome $Y$ is given by $\Delta(\boldsymbol{X})$ and $\Upsilon(\boldsymbol{X})$. Following this model, $\Gamma(\boldsymbol{X})$, $\Delta(\boldsymbol{X})$, and $\Upsilon(\boldsymbol{X})$ are formulated as three neural network encoders, each of which extracts the representation of *instrumental variables* (i.e., the features that influence $A$ but not $Y$), confounders, and adjustment variables, respectively. For example, in medical treatment setups, a possible instrumental

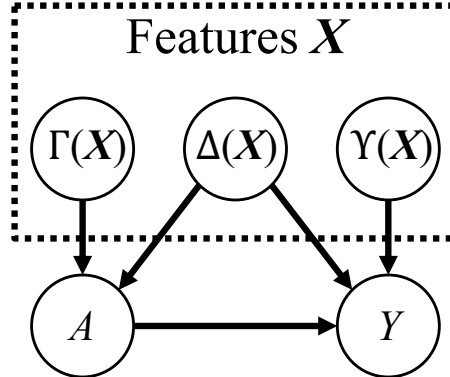

Figure 1: Graphical model illustration of DRCFR method

variable, confounder, and adjustment variable might be income, age, and smoking habits, respectively.

To learn the representations of such features, DRCFR minimizes the weighted loss by computing the weights based on the inverse of propensity scores, given by $\pi(\Gamma(X), \Delta(X))$.

A strong advantage of DRCFR is that it can avoid losing the information of adjustment variables, represented by $\Upsilon(X)$, which is useful for outcome prediction. However, since the inverse of conditional probability, $\pi(\Gamma(X), \Delta(X))$, often yields extreme values, under finite sample settings, even a slight estimation error of $\pi$ leads to a large weight estimation error. This numerical instability of weight estimation makes it difficult to achieve high CATE estimation performance.

## 3 PROPOSED METHOD

To improve the estimation stability of weighted representation learning, we propose a differentiable weight correction framework that can be used in an end-to-end fashion.

### 3.1 OVERVIEW

To estimate the CATE in (1), following DRCFR [Hassanpour and Greiner, 2020], we perform weighted representation learning. We learn three model components: the feature representations (i.e., $\Gamma(X)$, $\Delta(X)$, and $\Upsilon(X)$ in Figure 1), propensity score model $\pi(\Gamma(X), \Delta(X))$, and outcome prediction models $h^0(\Delta(X), \Upsilon(X))$ and $h^1(\Delta(X), \Upsilon(X))$, where $h^0$ and $h^1$ are used to predict potential outcomes $Y^0$ and $Y^1$.

DRCFR jointly optimizes these three model components by minimizing the weighted loss. However, we empirically observed that such a joint optimization is difficult. A possible reason is that the loss function dramatically changes with the IPW weights and hence substantially varies with the parameter values of propensity score $\pi$. For this reason, we separately learn $\pi$ and perform an alternate optimization that repeatedly takes the following two steps.

First, we learn propensity score $\pi$ (while fixing the other model parameters) by minimizing the cross entropy loss:

$$\min_{\pi} -\frac{1}{n} \sum_{i=1}^{n} \Big( a_i \log(\pi(\Gamma(\boldsymbol{x}_i), \Delta(\boldsymbol{x}_i)))$$
$$+ (1 - a_i) \log(1 - \pi(\Gamma(\boldsymbol{x}_i), \Delta(\boldsymbol{x}_i))) \Big) + \lambda_\pi \Omega(\pi), \quad (2)$$

where $\Omega(\cdot)$ is a regularizer that penalizes the model complexity, and $\lambda_\pi > 0$ is a regularization parameter.

Second, we learn the other model parameters (with $\pi$'s parameters fixed) by minimizing the weighted loss:

$$\min_{\Gamma, \Delta, \Upsilon, h^0, h^1} \frac{1}{n} \sum_{i=1}^{n} w_i l(y_i, h^{a_i}(\Delta(\boldsymbol{x}_i), \Upsilon(\boldsymbol{x}_i)))$$
$$+ \lambda_\Upsilon \text{MMD}\Big(\{\Upsilon(\boldsymbol{x}_i)\}_{i:a_i=0}, \{\Upsilon(\boldsymbol{x}_i)\}_{i:a_i=1}\Big)$$
$$+ \lambda_{-\pi} \Omega\Big(\Gamma, \Delta, \Upsilon, h^0, h^1\Big), \quad (3)$$

where $w_i$ is the weight that is given by propensity score $\pi(\Gamma(\boldsymbol{x}_i), \Delta(\boldsymbol{x}_i))$, $l$ is the prediction loss for outcome $y_i$, $\lambda_\Upsilon > 0$ and $\lambda_{-\pi} > 0$ are regularization parameters,[1] and MMD denotes the kernel maximum mean discrepancy (MMD) [Gretton et al., 2012], which measures the discrepancy between empirical conditional distributions $\hat{P}(\Upsilon(X) \mid A = 0)$ and $\hat{P}(\Upsilon(X) \mid A = 1)$. Regularizing this MMD term prohibits $\Upsilon$ from having any information about treatment $A$, thus making $\Upsilon(X)$ a good representation of the adjustment variables.

To achieve a high CATE estimation performance, how to compute the weight value (i.e., $w_i$ in (3)) is essential. In the DRCFR method [Hassanpour and Greiner, 2020], weight is formulated using *importance sampling*, which employs a density-ratio-based weight to construct a weighted estimator of expected value. To estimate the expected outcome prediction losses over the observed individuals ($A = a_i$) and the unobserved individuals ($A = 1 - a_i$), DRCFR formulates the weight as the sum of two density ratios:

$$w_i = \frac{P(\Gamma(\boldsymbol{x}_i), \Delta(\boldsymbol{x}_i) \mid A = a_i)}{P(\Gamma(\boldsymbol{x}_i), \Delta(\boldsymbol{x}_i) \mid A = a_i)} + \frac{P(\Gamma(\boldsymbol{x}_i), \Delta(\boldsymbol{x}_i) \mid A = 1 - a_i)}{P(\Gamma(\boldsymbol{x}_i), \Delta(\boldsymbol{x}_i) \mid A = a_i)}$$
$$= 1 + \frac{P(A = 1 - a_i \mid \Gamma(\boldsymbol{x}_i), \Delta(\boldsymbol{x}_i))}{P(A = 1 - a_i)} \frac{P(A = a_i)}{P(A = a_i \mid \Gamma(\boldsymbol{x}_i), \Delta(\boldsymbol{x}_i))}$$
$$= 1 + \frac{P(A = a_i)}{P(A = 1 - a_i)} \left( \frac{1}{P(A = a_i \mid \Gamma(\boldsymbol{x}_i), \Delta(\boldsymbol{x}_i))} - 1 \right) \quad (4)$$
$$\propto \frac{1}{P(A = a_i \mid \Gamma(\boldsymbol{x}_i), \Delta(\boldsymbol{x}_i))} := \frac{1}{\pi_{a_i}(\Gamma(\boldsymbol{x}_i), \Delta(\boldsymbol{x}_i))},$$

where $\pi_{a_i}(X) = a_i \pi(X) + (1 - a_i)(1 - \pi(X))$. Here weight $w_i$ is proportional to the inverse of propensity score $\pi_{a_i}(\Gamma(\boldsymbol{x}_i), \Delta(\boldsymbol{x}_i))$. Since such an IPW weight often takes an extreme value, the weight estimation is numerically unstable, leading to inaccurate CATE estimation. This issue is serious in a high-dimensional setup due to the difficulty of correctly estimating propensity scores [Assaad et al., 2021].

---

[1] $-\pi$ denotes the other model components than $\pi$.

To resolve this issue, we improve the weight stability by replacing an extreme value of $w_i$ in Eq. (4) with a weight stabilization technique, called Pareto smoothing.

## 3.2 WEIGHT CORRECTION VIA PARETO SMOOTHING

Pareto smoothing [Vehtari et al., 2024] is a technique for improving the weight stability of importance sampling.

According to Vehtari et al. [2024], this technique has two advantages. First, it can yield a less biased estimator, compared with weight truncation, which replaces extreme weights naively with constants [Crump et al., 2009, Ionides, 2008]:

$$w_i^{\text{Trunc.}} := \begin{cases} L & \text{if} & w_i < L \\ w_i & \text{if} & L \le w_i < U \\ U & \text{if} & U \le w_i \end{cases}, \qquad (5)$$

where $L > 0$ and $U > 0$ are the truncation thresholds. Second, it can be combined with self-normalization, which prevents the weights from being too small or too large relative to each other by dividing each weight value by its empirical mean under identical treatment assignment:

$$w_i^{\text{Norm.}} := \frac{w_i}{\overline{w^{A=a_i}}}, \quad \text{where} \quad \overline{w^{A=a_i}} = \frac{\sum_{j=1}^{n} \mathbf{I}(a_j = a_i) w_j}{\sum_{j=1}^{n} \mathbf{I}(a_j = a_i)}. \quad (6)$$

Here $\mathbf{I}(a_j = a_i)$ is an indicator function that takes 1 if $a_j = a_i$; otherwise, 0. We experimentally confirmed that performing self-normalization over Pareto-smoothed weights leads to better CATE estimation performance (Section 4.1).

To construct a weighted estimator that is numerically robust to weight estimation error, Pareto smoothing replaces the extremely large weight values with GPD quantiles in two steps: GPD parameter estimation and weight replacement.

### 3.2.1 GPD Parameter Fitting

First, we fit the GPD parameters to large IPW weight values.

Suppose that random variable $W$ follows the GPD. Then its GPD cumulative distribution function is defined as

$$F(w) = \begin{cases} 1 - \left(1 + \frac{\xi(w-\mu)}{\sigma}\right)^{-\frac{1}{\xi}} & (\xi \ne 0), \\ 1 - e^{-\frac{w-\mu}{\sigma}} & (\xi = 0) \end{cases} \qquad (7)$$

where $\mu \in \mathbb{R}$, $\sigma > 0$, and $\xi \in \mathbb{R}$ are location, scale, and shape parameters.

We fit these GPD parameters to the $M + 1$ largest IPW weight values. Here $M$ ($0 < M < n$) is given by heuristics; following Vehtari et al. [2024], we determine it by

$$M = \min \left\{ \left\lfloor \frac{n}{5} \right\rfloor, \lfloor 3 \sqrt{n} \rfloor \right\}, \qquad (8)$$

where $\lfloor n \rfloor$ denotes a floor function, which returns the greatest integer that is less than or equal to $n$. Letting $w_{[1]} \le \cdots \le w_{[n]}$ be the weights sorted in ascending order, the $M + 1$ largest ones are denoted by $w_{[n-M]}, \ldots, w_{[n]}$.

Following Vehtari et al. [2024], we set location parameter $\mu$ to the $(M + 1)$-th largest IPW weight value, i.e.,

$$\hat{\mu} = w_{[n-M]}. \qquad (9)$$

By contrast, we estimate $\sigma$ and $\xi$ using $w_{[n-M+1]}, \ldots, w_{[n]}$. Among several estimators, we employ the standard method termed the probability weighted moment (PWM) [Hosking and Wallis, 1987], [2] which constructs the estimators of $\sigma$ and $\xi$ using the following weighted moment statistic:

$$\alpha_s = \mathbb{E}\left[(1 - F(W))^s (W - \mu)\right] \quad s \in \{0, 1\}. \qquad (10)$$

Roughly speaking, statistic $\alpha_s$ in (10) is a weighted average of $W - \mu$ with weight $(1 - F(W))^s$ and is estimated as

$$\hat{\alpha}_0 = \frac{1}{M} \sum_{i=n-M+1}^{n} (w_{[i]} - \hat{\mu}), \qquad (11)$$

$$\hat{\alpha}_1 = \frac{1}{M} \sum_{i=n-M+1}^{n} (n - i)(w_{[i]} - \hat{\mu}). \qquad (12)$$

Using $\hat{\alpha}_0$ and $\hat{\alpha}_1$, the PWM method estimates $\sigma$ and $\xi$ as

$$\hat{\sigma} = \frac{2\hat{\alpha}_0 \hat{\alpha}_1}{\hat{\alpha}_0 - 2\hat{\alpha}_1}, \qquad (13)$$

$$\hat{\xi} = 2 - \frac{\hat{\alpha}_0}{\hat{\alpha}_0 - 2\hat{\alpha}_1}. \qquad (14)$$

### 3.2.2 Weight Replacement with GPD Quantiles

Second, we replace the $M$ largest weight values, i.e., $w_{[n-M+1]}, \ldots, w_{[n]}$, with the quantiles of the fitted GPD with parameters $(\hat{\mu}, \hat{\sigma}, \hat{\xi})$ in (9), (13), and (14).

Since the quantile function is given by the inverse of the cumulative distribution function, we replace weight value $w_{[n-M+m]}$ ($m = 1, \ldots, M$) with $\frac{m-1/2}{M}$-quantile as

$$w_{[n-M+m]} = \hat{F}^{-1}\left(\frac{m - 1/2}{M}\right), \qquad (15)$$

where $\hat{F}$ denotes a fitted GPD cumulative distribution function. By contrast, we do not change the other weight values, i.e., $w_{[1]}, \ldots, w_{[n-M]}$. Hence, letting $i = n - M + m$, we can summarize the weight replacement procedure as

$$w_{[i]} = \mathbf{I}(i \ge n - M + 1) \hat{F}^{-1}\left(\frac{i - (n - M) - 1/2}{M}\right)$$
$$+ (1 - \mathbf{I}(i \ge n - M + 1)) w_{[i]}. \qquad (16)$$

---

[2] Although we empirically observed that using PWM leads to good CATE estimation performance, GPD parameter fitting might not be easy in general. However, according to Vehtari et al. [2024, Section 6], one can evaluate the reliability of GPD fitting by employing the estimated value of GPD's shape parameter, $\hat{\xi}$, which determines the heaviness of the distribution tail.

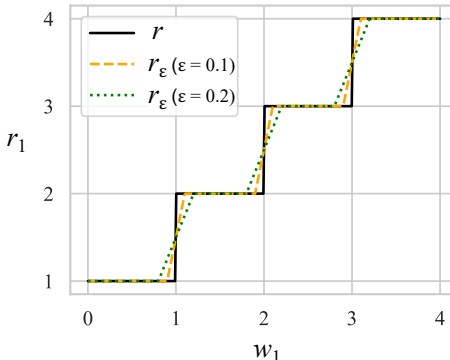

Figure 2: Illustration of rank function $r = r(w)$ (black) and differentiable rank functions $r = r_\varepsilon(w)$ (orange and green): Here we take input vector $w = [w_1, 1, 2, 3]^\top$, vary $w_1$'s value and look at how its rank $r_1 \in r$ changes. When regularization parameter $\varepsilon \to 0$, $r_\varepsilon$ converges to $r$ [Blondel et al., 2020].

In this paper, we utilize the weight replacement formula in (16) to improve the estimation stability of weighted representation learning. Unfortunately, we cannot directly employ this formula in an end-to-end manner because it needs non-differentiable computations.

## 3.3 NON-DIFFERENTIABLE PROCEDURES

The main difficulty of using Pareto smoothing for weighted representation learning is that it requires the computation of the *rank* of each IPW weight.

Ranking is an operation that takes input vector $w = [w_1, \ldots, w_n]^\top$ and outputs the position of each element $w_i$ in sorted vector $[w_{[1]}, \ldots, w_{[n]}]^\top$, where $w_{[1]} \leq \cdots \leq w_{[n]}$. To illustrate this operation, consider a case with $n = 3$. For instance, if $w$ satisfies $w_3 \leq w_1 \leq w_2$, since $w_1 = w_{[2]}$, $w_2 = w_{[3]}$, and $w_3 = w_{[1]}$ hold, the rank of $w$ is given as vector $r = [2, 3, 1]^\top$. Formally, such an operation can be expressed as $r = r(w)$, using function $r$, called a rank function (See Appendix A for the definition of function $r$).

Unfortunately, this rank function is not differentiable with respect to input $w$. To see this, consider $w = [w_1, 1, 2, 3]^\top$ and observe how the rank of $w_1$ varies when we increase its value. In this case, its rank, $r_1$, is given as a piecewise constant function, as illustrated as the black line in Figure 2. Since the derivative of such a piecewise constant function is **always** zero or undefined, we cannot perform gradient backpropagation and hence cannot employ the weight correction technique in (16) in an end-to-end manner. Therefore, with such a non-differentiable rank function, we cannot use Pareto smoothing for weighted representation learning, which jointly learns the propensity score model, the feature representations, and the outcome prediction models.

One may consider a separate learning approach that trains the propensity score model, computes the Pareto-smoothed

IPW weights by (16), and learns the feature representations. This approach, however, requires directly fitting a propensity score model to features $X$, not their representations. Since accurately estimating a propensity score model from high-dimensional features $X$ is considerably difficult, such a separate learning approach yields large model misspecification error and hence leads to CATE estimation bias. We experimentally show its poor performance in Section 4.1.

For this reason, we develop a joint learning approach by making the non-differentiable computation in Pareto smoothing differentiable.

## 3.4 MAKING PARETO SMOOTHING DIFFERENTIABLE

### 3.4.1 Differentiable Approximation

The weight replacement formula in (16) requires the computation of two troublesome piecewise constant functions. One is rank function $r$, which is needed to obtain the position of weight $w_i$ in sorted vector $[w_{[1]}, \ldots, w_{[n]}]^\top$, and the other is indicator function $\mathbf{I}(i \geq n - M + 1)$.

To make rank function $r$ differentiable, we utilize the differentiable ranking technique [Cuturi et al., 2019, Blondel et al., 2020], which approximates rank function $r(w)$ with a differentiable function. Among the recent methods, we select a computationally efficient one [Blondel et al., 2020], which works with $O(n\log n)$ time and $O(n)$ memory complexity. With this method, we approximate rank function $r(w)$ as the solution to the regularized linear programming (LP) that contains the $l^2$ regularization term with regularization parameter $\varepsilon > 0$. The solution, $r_\varepsilon(w)$, is a piecewise linear function that can well approximate rank function $r$ (as illustrated in Figure 2) and is differentiable almost everywhere, thus greatly facilitating gradient backpropagation.

As a differentiable approximation of indicator function $\mathbf{I}$ in (16), we employ sigmoid function $\varsigma$:

$$\mathbf{I}(i \geq j) \simeq \varsigma(i, j) := \frac{1}{1 + e^{-\kappa(i-j)}}, \qquad (17)$$

where $\kappa > 0$ is a hyperparameter.

### 3.4.2 Reformulation of GPD Parameter Estimators

To employ differentiable rank $r = r_\varepsilon(w)$ for Pareto smoothing, since it represents ranks as continuous values, we need to modify the GPD parameter estimators, i.e., $\hat{\mu}$, $\hat{\sigma}$, and $\hat{\xi}$.

Regarding location parameter $\hat{\mu}$ in (9), since this estimator is given as $w_{[n-M]}$, i.e., the largest weight among $w_{[1]}, \ldots, w_{[n-M]}$, we reformulate it as

$$\tilde{\mu} = w_i, \quad \text{where } i = \arg\max_i \{r_i \mid r_i \leq n - M\}.$$

To reformulate $\hat{\sigma}$ and $\hat{\xi}$ in (13) and (14), we rephrase estimators $\hat{\alpha}_0$ and $\hat{\alpha}_1$ in (11) and (12). With non-differentiable rank $\boldsymbol{r} = r(\boldsymbol{w})$, these estimators are equivalently reformulated by rewriting the summation over $w_{[n-M+1]}, \ldots, w_{[n]}$ in (11) and (12) with indicator function $\mathbf{I}$ as

$$\hat{\alpha}_0 = \frac{1}{M} \sum_{i=1}^{n} \mathbf{I}(r_i \geq n - M + 1)\,(w_i - \hat{\mu})$$

$$\hat{\alpha}_1 = \frac{1}{M} \sum_{i=1}^{n} \mathbf{I}(r_i \geq n - M + 1)\,(n - r_i)\,(w_i - \hat{\mu}).$$

Hence, when given differentiable rank $\boldsymbol{r} = r_\varepsilon(\boldsymbol{w})$, by replacing indicator function $\mathbf{I}$ with sigmoid function $\varsigma$ in (17), we make $\hat{\alpha}_0$ and $\hat{\alpha}_1$ differentiable with respect to $\boldsymbol{r}$:

$$\tilde{\alpha}_0 = \frac{1}{\tilde{M}} \sum_{i=1}^{n} \varsigma(r_i, n - M + 1)\,(w_i - \tilde{\mu}), \qquad (18)$$

$$\tilde{\alpha}_1 = \frac{1}{\tilde{M}} \sum_{i=1}^{n} \varsigma(r_i, n - M + 1)\,(n - r_i)\,(w_i - \tilde{\mu}), \qquad (19)$$

where $\tilde{M} = \sum_{i=1}^{n} \varsigma(r_i, n - M + 1)$. By substituting $\tilde{\alpha}_0$ and $\tilde{\alpha}_1$ for $\hat{\alpha}_0$ and $\hat{\alpha}_1$ in (13) and (14), we compute scale and shape parameters as $\tilde{\sigma}$ and $\tilde{\xi}$, respectively.

### 3.4.3 Overall Algorithm

Using the GPD cumulative distribution function, $\tilde{F}$, with parameters $(\tilde{\mu}, \tilde{\sigma}, \tilde{\xi})$, we replace each weight $w_i$ in (4) with

$$\tilde{w}_i = \varsigma(r_i, n - M + 1)\,\tilde{F}^{-1}\left(\zeta\left(\frac{r_i - (n - M) - 1/2}{M}\right)\right)$$
$$+ (1 - \varsigma(r_i, n - M + 1))\,w_i, \qquad (20)$$

where $\zeta(x) := \min\{\max\{x, 0\}, 1\}$ is a function that forces input $x$ to lie in $[0, 1]$. Using Pareto-smoothed weight $\tilde{w}_i$ instead of $w_i$, we minimize the objective function in (3).

Algorithm 1 summarizes our method. To alternately minimize the objective functions in (2) and (3), we perform stochastic gradient descent [Kingma and Ba, 2015]. After the convergence, we estimate the CATE in (1) by $h^1(\Delta(\boldsymbol{x}), \Upsilon(\boldsymbol{x})) - h^0(\Delta(\boldsymbol{x}), \Upsilon(\boldsymbol{x}))$.

Compared with the DRCFR method [Hassanpour and Greiner, 2020], our method requires additional time to compute Pareto-smoothed weights (lines 12-16 in Algorithm 1). In particular, computing differentiable rank (line 12) requires time complexity $O(B \log B)$ for mini-batch size $B$, which is needed to evaluate the objective function in (3) and its gradient for each iteration in the training phase.

**Remark**: Strictly speaking, the choice of activation functions in propensity score $\pi$ and feature representations $\Gamma$ and $\Delta$ is critical for satisfying the assumption of Pareto smoothing that the distribution of the importance sampling weight

---

**Algorithm 1** Differentiable Pareto-Smoothed Weighting (DPSW)

1: Initialize the parameters of $\Gamma, \Delta, \Upsilon, \pi, h^0$, and $h^1$
2: **while** not converged **do**
3:     **while** not converged **do**
4:         Sample mini-batch from $\mathcal{D} = \{(a_i, \boldsymbol{x}_i, y_i)\}_{i=1}^{n}$
5:         Update $\pi$ by minimizing cross entropy loss in (2)
6:     **end while**
7:     **while** not converged **do**
8:         Sample mini-batch from $\mathcal{D} = \{(a_i, \boldsymbol{x}_i, y_i)\}_{i=1}^{n}$
9:         **for** instance $i$ in mini-batch **do**
10:           Compute weight $w_i$ by (4)
11:         **end for**
12:     Compute differentiable rank $\boldsymbol{r} = r_\varepsilon(\boldsymbol{w})$
13:     Estimate GPD parameters as $\tilde{\mu}, \tilde{\sigma}$, and $\tilde{\xi}$
14:     **for** instance $i$ in mini-batch **do**
15:         Replace each weight $w_i$ with $\tilde{w}_i$ in (20)
16:     **end for**
17:     Update $\Gamma, \Delta, \Upsilon, h^0$, and $h^1$ by minimizing prediction loss in (3) with Pareto-smoothed weights $\{\tilde{w}_i\}$
18:     **end while**
19: **end while**

---

is absolutely continuous, which is necessary to prove the asymptotic consistency (Theorem 1 of Vehtari et al. [2024]). This assumption holds if each activation is differentiable almost everywhere (i.e., differentiable except on a set of measure zero). However, for instance, using the rectified linear unit (ReLU) in propensity score model $\pi$ makes the distribution of IPW weight discontinuous, thus violating the assumption of Pareto smoothing. Even with almost everywhere differentiable activation functions, due to the lack of learning theory on neural network models, deriving the asymptotic consistency of our CATE estimator is extremely challenging and is left as our future work.

## 4 EXPERIMENTS

### 4.1 SEMI-SYNTHETIC DATA

First, we evaluated the CATE estimation performance using semi-synthetic benchmark datasets, where the true CATE values are available, unlike real-world data.

**Data:** We selected the two high-dimensional datasets: the News and the Atlantic Causal Inference Conference (ACIC) datasets [Johansson et al., 2016, Shimoni et al., 2018].

The News dataset is constructed from $n = 5000$ articles, randomly sampled from the New York Times corpus in the UCI repository. [3] The task is to infer the effect of the viewing device (desktop ($A = 0$) or mobile ($A = 1$)) on the readers' experience $Y$. Features $X$ are the count of $d = 3477$ words in

---

[3]https://archive.ics.uci.edu/dataset/164/bag+of+words

each article. Treatment $A$ and outcome $Y$ are simulated using the latent topic variables obtained by fitting a topic model on $X$. The ACIC dataset is derived from the clinical measurements of $d = 177$ features in the Linked Birth and Infant Death Data (LBIDD) [MacDorman and Atkinson, 1998], which was developed for a data analysis competition called ACIC2018. We randomly selected $n = 5000$ observations and prepared 20 datasets. With both semi-synthetic datasets, we randomly split each sample into training, validation, and test data with a ratio of 60/20/20.

**Baselines:** To evaluate our method (**DPSW**) and its variant that performs self-normalization (**DPSW Norm.**), we consider 15 baselines. With DRCFR [Hassanpour and Greiner, 2020], we tested four different weighting schemes: no weight modification (**DRCFR**), self-normalization (**DRCFR Norm.**; Eq. (6)), weight truncation with a threshold suggested by Crump et al. [2009] (**DRCFR Trunc.**; Eq. (5)), and a scheme that ignores the prediction loss for individuals with extreme weights based on the same threshold (**DRCFR Ignore**). We also tested a separate learning approach (**PSW**; Section 3.3), which trains propensity score $\pi$ with $\{(a_i, x_i)\}_{i=1}^n$ beforehand and learns only $\Delta(X)$ and $\Upsilon(X)$ using Pareto-smoothed IPW weights. Other baselines include (i) linear regression methods: a single model with treatment $A$ added to its input (**LR-1**) and two separate models for each treatment (**LR-2**); (ii) meta-learner methods: the S-Learner (**SL**), the T-Learner (**TL**), the X-Learner (**XL**), and the DR-Learner (**DRL**); (iii) tree-based methods: causal forest [Athey et al., 2019] (**CF**) and a variant combined with double/debiased machine learning [Chernozhukov et al., 2018] (**CF DML**); and (iv) neural network methods: the treatment-agnostic regression network [Shalit et al., 2017] (**TARNet**) and the generative adversarial network [Yoon et al., 2018] (**GANITE**). We describe the settings of these baselines in Appendix B.1.

**Settings:** Regarding our method and DRCFR, we used three-layered feed-forward neural networks (FNNs) to formulate feature representations $\Gamma(X)$, $\Delta(X)$, and $\Upsilon(X)$, propensity score $\pi$, and outcome prediction models $h^0$ and $h^1$.

We tuned the hyperparameters (e.g., parameter $\varepsilon$ of differentiable rank $r_\varepsilon(w)$ in our method) by minimizing the objective function value on the validation data; such hyperparameter tuning is standard for CATE estimation [Shalit et al., 2017].

**Performance metric:** Following Hill [2011], we used the precision in the estimation of heterogeneous effect (PEHE), PEHE $:= \sqrt{\frac{1}{n}\left((y_i^1 - y_i^0) - \hat{\tau}_i\right)^2}$, where $y_i^0$ and $y_i^1$ are the true potential outcomes, and $\hat{\tau}_i$ denotes the predicted CATE value. We computed the mean and the standard deviation of the test PEHEs over 50 realizations of potential outcomes (the News dataset) and 20 realizations (the ACIC dataset).

**Results:** Table 1 presents the test PEHEs on the News and ACIC datasets.

Table 1: Mean and standard deviation of test PEHE on semi-synthetic datasets (lower is better)

| Method | News ($d = 3477$) | ACIC ($d = 177$) |
|---|---|---|
| LR-1 | $3.35 \pm 1.28$ | $0.72 \pm 0.07$ |
| LR-2 | $5.36 \pm 1.75$ | $3.82 \pm 0.15$ |
| SL | $2.83 \pm 1.11$ | $1.69 \pm 0.52$ |
| TL | $2.55 \pm 0.82$ | $2.23 \pm 0.50$ |
| XL | $2.77 \pm 1.01$ | $1.05 \pm 0.72$ |
| DRL | $23.9 \pm 5.96$ | $3.77 \pm 8.96$ |
| CF | $3.84 \pm 1.67$ | $3.55 \pm 0.19$ |
| CF DML | $2.69 \pm 1.06$ | $1.18 \pm 0.32$ |
| TARNet | $4.92 \pm 1.80$ | $3.31 \pm 0.51$ |
| GANITE | $2.68 \pm 0.66$ | $3.69 \pm 0.77$ |
| DRCFR | $2.38 \pm 0.66$ | $0.98 \pm 0.07$ |
| DRCFR Norm. | $2.37 \pm 0.94$ | $0.73 \pm 0.12$ |
| DRCFR Trunc. | $2.42 \pm 0.79$ | $1.06 \pm 0.06$ |
| DRCFR Ignore | $2.35 \pm 0.75$ | $0.84 \pm 0.06$ |
| PSW | $4.03 \pm 1.35$ | $0.71 \pm 0.01$ |
| **DPSW** | $\mathbf{2.20 \pm 0.72}$ | $\mathbf{0.57 \pm 0.03}$ |
| **DPSW Norm.** | $\mathbf{2.10 \pm 0.66}$ | $\mathbf{0.52 \pm 0.16}$ |

Our proposed frameworks (**DPSW** and **DPSW Norm.**) outperformed all the baselines, demonstrating their effectiveness in CATE estimation from high-dimensional data. **DPSW Norm.** achieved lower PEHEs than **DPSW**, implying that the self-normalization of Pareto-smoothed weights further improves the stability of the weight estimation.

Weighted representation learning methods (DR-CFR and DPSW) outperformed the other neural network methods (TARNet and GANITE), especially on the ACIC dataset. Given that treatment $A$ and outcome $Y$ of this dataset were simulated using different features in $X$, these results emphasize the importance of performing data-driven feature separation by weighted representation in such a setup.

PSW worked much worse on the News dataset than DPSW, indicating that fitting a propensity score directly to high-dimensional features $X$ leads to severe model misspecification error, complicating subsequent weighted representation learning, even with Pareto-smoothed weights. By contrast, our joint learning approach performed well, since it used differentiable Pareto smoothing.

## 4.2 SYNTHETIC DATA

Next we investigated how well our method learned the feature representations using synthetic data, where the data-generating processes are entirely known.

**Data:** Following Hassanpour and Greiner [2020], we simulated the synthetic data. We randomly generated features

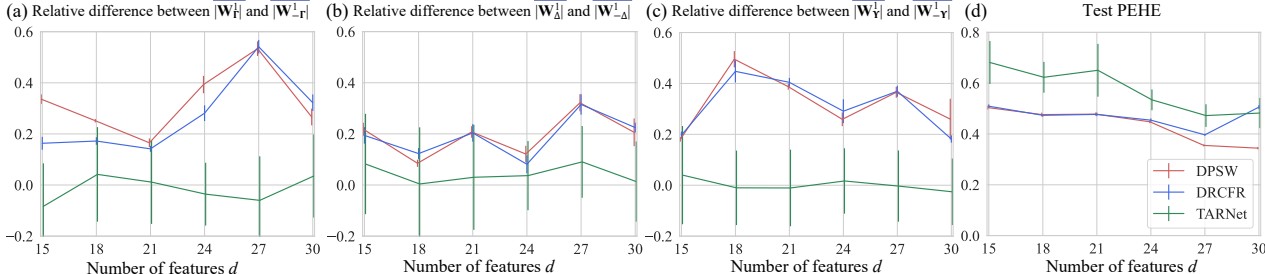

Figure 3: Learned encoder parameter differences and test PEHEs on synthetic data: (a): value difference of $\mathbf{W}^1$ in encoder $\Gamma(X)$; (b): value difference of $\mathbf{W}^1$ in encoder $\Delta(X)$; (c): value difference of $\mathbf{W}^1$ in encoder $\Upsilon(X)$; (d) test PEHEs. With TARNet, since it learns a single encoder, we computed all parameter value differences with weight matrix in same encoder.

$X = [X_\Gamma, X_\Delta, X_\Upsilon]^\top \in \mathbb{R}^d$ ($d = 15, 18, \ldots, 30$). Next, by regarding feature subsets $X_\Gamma \in \mathbb{R}^{d/3}$, $X_\Delta \in \mathbb{R}^{d/3}$, and $X_\Upsilon \in \mathbb{R}^{d/3}$ as instrumental variables, confounders, and adjustment variables, respectively, we sampled binary treatment $A$ using $X_\Gamma$ and $X_\Delta$ and outcome $Y$ by employing $X_\Delta$ and $X_\Upsilon$ (See Appendix B.2 for details). We split each of the 20 datasets ($n = 20000$) with a 50/25/25 training/validation/test ratio.

**Performance metric:** As with Hassanpour and Greiner [2020], we evaluated the quality of the learned feature representations $\Gamma(X)$, $\Delta(X)$, and $\Upsilon(X)$, each of which is formulated as a three-layered FNN encoder:

$$\text{FNN}(X) \coloneqq \nu\left(\mathbf{W}^3 \nu\left(\mathbf{W}^2 \nu\left(\mathbf{W}^1 X\right)\right)\right),$$

where $\nu$ is the exponential linear units (ELUs) [Clevert et al., 2016], and $\mathbf{W}^1$, $\mathbf{W}^2$, and $\mathbf{W}^3$ are the weight parameter matrices in the first, second, and third layer, respectively.

To determine whether the learned FNN encoders correctly look at important features, we measured the attribution of features $X_\Gamma$, $X_\Delta$, and $X_\Upsilon$ by employing $\mathbf{W}^1$, i.e., the trained weight matrix in the first layer of each encoder. For instance, we quantified $X_\Gamma$'s attribution on $\Gamma(X)$ in two steps. First, we partitioned its learned weight parameter matrix as $\mathbf{W}^1 = [\mathbf{W}^1_\Gamma, \mathbf{W}^1_{-\Gamma}]$, where $\mathbf{W}^1_\Gamma$ is a submatrix with the first $d/3$ columns of $\mathbf{W}^1$ and $\mathbf{W}^1_{-\Gamma}$ is the one with the other columns. Then we measured how greatly features $X_\Gamma$ affect learned representation $\Gamma(X)$ by taking the relative difference between the average absolute values of the weight parameter submatrices, i.e., $\frac{\overline{|\mathbf{W}^1_\Gamma|} - \overline{|\mathbf{W}^1_{-\Gamma}|}}{\overline{|\mathbf{W}^1_{-\Gamma}|}}$. We evaluated other learned representations, $\Delta(X)$ and $\Upsilon(X)$, in the same way.

**Results:** Figure 3 shows the means and standard deviations of the learned parameter differences and the test PEHEs over 20 randomly generated synthetic datasets.

With our DPSW method and DRCFR, absolute parameter values $|\mathbf{W}^1_\Gamma|$, $|\mathbf{W}^1_\Delta|$, and $|\mathbf{W}^1_\Gamma|$ were sufficiently larger than $|\mathbf{W}^1_{-\Gamma}|$, $|\mathbf{W}^1_{-\Delta}|$, and $|\mathbf{W}^1_{-\Gamma}|$, respectively, showing that both methods correctly learned $\Gamma(X)$, $\Delta(X)$, and $\Upsilon(X)$ that are highly dependent on instrumental variables $X_\Gamma$, confounders

$X_\Delta$, and adjustment variables $X_\Upsilon$, respectively. These results offer a clear contrast to TARNet, which learns a single representation without feature separation.

The same is true for the CATE estimation performance (Figure 3 (d)). TARNet's test PEHE was larger than DPSW and DRCFR, demonstrating the importance of data-driven feature separation by weighted representation learning. By contrast, our method achieved the lowest PEHE, thus indicating that our weight correction framework successfully improved the CATE estimation performance of DRCFR.

**Performance under high-dimensional setup:** We confirmed that our method also worked well with $d = 600, 1200, \ldots, 3000$ (See Appendix C for details).

## 5 RELATED WORK

**Data-driven feature separation for CATE estimation:** CATE estimation has gained increasing attention because of its great importance for causal mechanism understanding [Chikahara et al., 2022, Zhao et al., 2022] and for decision support in various fields, such as precision medicine [Gao et al., 2021] and online advertising [Sun et al., 2015]. There has been a surge of interest in leveraging flexible machine learning models, including tree-based models [Athey et al., 2019, Hill, 2011], Gaussian processes [Alaa and van der Schaar, 2018, Horii and Chikahara, 2024], and neural networks [Hassanpour and Greiner, 2019, Johansson et al., 2016, Shalit et al., 2017]. However, most methods treat all input features $X$ as confounders. As pointed out by Wu et al. [2022], the empirical performance of such methods greatly varies with the presence of adjustment variables in $X$, which is usual in practice, especially in high-dimensional settings.

Motivated by this issue, we develop data-driven feature separation methods for treatment effect estimation. A pioneering work is data-driven variable decomposition (D²VD) [Kuang et al., 2017, 2020], which minimizes the weighted prediction loss plus the regularizer for feature separation. A recent method addresses a more complicated setup, where features $X$ include *post-treatment variables*, which are affected by

treatment $A$ [Wang et al., 2023]. However, the estimation target of these methods is ATE, not CATE.

By contrast, DRCFR deals with CATE estimation and is founded on weighted representation learning, which is a promising approach for addressing high-dimensional data. These advantages are why we adopted it as the inference engine of our weight correction framework. Integrating the recent idea of enforcing independence between feature representations with mutual information [Cheng et al., 2022, Chu et al., 2022, Liu et al., 2024] remains our future work.

**Weighting schemes for treatment effect estimation:** IPW [Rosenbaum and Rubin, 1983] is a common weighting technique for treatment effect estimation. However, a weighted estimator based on IPW is often numerically unstable due to the computation of the inverse of propensity scores. One remedy is weight truncation [Crump et al., 2009], which, however, causes estimation bias, leading to inaccurate treatment effect estimation.

To improve the estimation performance, Zhu et al. [2020] employed Pareto smoothing [Vehtari et al., 2024]. Although they empirically show that using this technique leads to better performance than weight truncation, their method was developed to estimate ATE, not CATE.

Applying Pareto smoothing in weighted representation learning for CATE estimation is difficult because it prevents gradient backpropagation due to non-differentiability. This difficulty is disappointing, given that previous work has theoretically shown that such weight correction schemes as weight truncation help extract predictive feature representations for CATE estimation [Assaad et al., 2021].

To establish a Pareto-smoothed weighting framework for CATE estimation from high-dimensional data, we demonstrated how a differentiable ranking technique [Blondel et al., 2020] can be used to simultaneously learn a propensity score model and feature representations.

# 6   CONCLUSION

We established a differentiable Pareto-smoothed weighting framework for CATE estimation from high-dimensional data. To construct a CATE estimator that is numerically robust to propensity score estimation error, we develop a differentiable weight correction procedure based on Pareto smoothing and incorporated it into weighted representation learning for CATE estimation. We experimentally show that our framework outperformed traditional weighting schemes as well as the existing CATE estimation methods.

By leveraging the versatility of weighting, our future work will investigate how to extend our framework to estimate the effects of high-dimensional binary treatment [Zou et al., 2020], continuous-valued treatment [Wang et al., 2022], and time series treatment [Lim et al., 2018].

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

# Supplementary Materials for
# "Differentiable Pareto-Smoothed Weighting
# for High-Dimensional Heterogeneous Treatment Effect Estimation"

**Yoichi Chikahara**[1]                    **Kansei Ushiyama**[2*]

[1]NTT Communication Science Laboratories, Kyoto, Japan
[2]The University of Tokyo, Tokyo, Japan

## A    RANK FUNCTION DEFINITION

In Section 3.3, we consider rank function $r$, which takes input vector $\boldsymbol{w} = [w_1, \ldots, w_n]^\top$ and outputs the rank of each element in $\boldsymbol{w}$. We formally define this rank function based on a sorting operation and the concept of inverse permutation.

Consider a sorting operation over $w_1, \ldots, w_n$ in $\boldsymbol{w} \in \mathbb{R}^n$ that finds permutation $\boldsymbol{\rho} = [\rho_1, \ldots, \rho_n]^\top$ such that the vector values that are permuted according to $\boldsymbol{\rho}$, $\boldsymbol{w_\rho} = [w_{\rho_1}, \ldots, w_{\rho_n}]^\top$, are increasing as $w_{\rho_1} \leq \cdots \leq w_{\rho_n}$. Let $\boldsymbol{\rho}^{-1}$ be the inverse of permutation $\boldsymbol{\rho}$, i.e., a permutation whose $\rho_i$-th element is $i$ for $i = 1, \ldots, n$.

Then the ranking function is defined as the inverse of the sorting permutation:

$$r(\boldsymbol{w}) = \boldsymbol{\rho}^{-1}(\boldsymbol{w}). \tag{21}$$

Throughout this paper, we consider rank function $r(\boldsymbol{w})$ that evaluates the position of each $w_i$ based on sorting in an ascending order. If we need to address the ranking in descending order, we can formulate it as $r(-\boldsymbol{w})$.

## B    EXPERIMENTAL DETAILS

### B.1    SETTINGS OF BASELINES

Regarding linear regression methods (LR-1 and LR-2), to avoid overfitting due to the large number of input features, we employed the ridge regression model in scikit-learn [Pedregosa et al., 2011]. For the meta-learner methods (i.e., the S-Learner (SL), the T-Learner (TL), the X-Learner (XL), the DR-Learner (DRL)) and the tree-based methods (CF and CF DML) we used the EconML Python package [Battocchi et al., 2019]. As the base learners for SL, TL, XL, and DRL, we chose random forest because we empirically observed that it achieved the best performance among the three model candidates: random forest, gradient boosting, and support vector machine.

We evaluated the performance of the neural network methods (TARNet and GANITE), with the existing implementations [Curth and van der Schaar, 2021, Yoon et al., 2018]. [1,2] By contrast, we implemented the DRCFR method using PyTorch [Paszke et al., 2019].

With PSW, we trained propensity score model $\pi(X)$ beforehand, using a paired sample $\{(a_i, \boldsymbol{x}_i)\} \subset \mathcal{D}$. We formulated $\pi(X)$ using a three-layered FNN, as with our method and DRCFR. After computing the IPW weights based on the trained propensity score model, we performed Pareto smoothing over them with the `psislw()` function in a Python package called ArviZ [Kumar et al., 2019]. Using the Pareto-smoothed weights, we learned two encoders $\Delta(X)$ and $\Upsilon(X)$, as well as outcome prediction models $h^0$ and $h^1$.

---

[1]https://github.com/AliciaCurth/CATENets
[2]https://github.com/jsyoon0823/GANITE

*Accepted for the 40th Conference on Uncertainty in Artificial Intelligence* (UAI 2024).

## B.2 SYNTHETIC DATA

Following Hassanpour and Greiner [2020], we prepared synthetic datasets.[3]

We first drew the values of features $X \in \mathbb{R}^d$ from standard multivariate Gaussian distribution:

$$X \sim \mathcal{N}(0, I), \tag{22}$$

where $\mathcal{N}$ denotes the Gaussian distribution.

Next, we sampled the values of treatment $A$ and outcome $Y$ using $X_\Gamma \in \mathbb{R}^{d/3}$, $X_\Delta \in \mathbb{R}^{d/3}$, and $X_\Upsilon \in \mathbb{R}^{d/3}$, which are the feature subsets in $X = [X_\Gamma, X_\Delta, X_\Upsilon]^\top$. In particular, we employed $\psi = [X_\Gamma, X_\Delta]^\top \in \mathbb{R}^{2d/3}$ to generate $A$'s values as

$$A \sim \text{Ber}\left(\frac{1}{1 + \exp(-c_A \cdot (\psi + 1))}\right), \tag{23}$$

where $c_A \in \mathbb{R}^{2d/3}$ is a coefficient vector drawn from $\mathcal{N}(0, 1)$, and $\mathbf{1} = [1, \ldots, 1]^\top$ is a vector with length $2d/3$. By contrast, we used $\phi = [X_\Delta, X_\Upsilon]^\top \in \mathbb{R}^{2d/3}$ to simulate potential outcomes $Y^0$ and $Y^1$ as

$$Y^0 = \frac{3}{2d} c_{Y^0} \cdot \phi + \epsilon \tag{24}$$

$$Y^1 = \frac{3}{2d} c_{Y^1} \cdot (\phi \odot \phi) + \epsilon, \tag{25}$$

$$\tag{26}$$

where $c_{Y^0} \in \mathbb{R}^{2d/3}$ and $c_{Y^1} \in \mathbb{R}^{2d/3}$ are the coefficient vectors drawn from $\mathcal{N}(0, 1)$, symbol $\odot$ denotes the element-wise product (a.k.a., Hadamard product), and $\epsilon \sim \mathcal{N}(0, 1)$ is scalar standard Gaussian noise. Using the values of potential outcomes $Y^0$ and $Y^1$, we computed the values of outcome $Y$ by $Y = AY^1 + (1 - A)Y^0$.

## C  ADDITIONAL EXPERIMENTAL RESULTS

In Section 4.2, as with Hassanpour and Greiner [2020], we present the performance on relatively low-dimensional synthetic data. To further investigate its performance, we ran additional experiments using high-dimensional synthetic data.

Here we generated the synthetic data in the same way (described in Appendix B.2) except that the number of features $d$ was set to $d = 600, 1200, \ldots, 3000$.

Figure 4 shows the results. Again, our method achieved better CATE estimation performance than DRCFR and TARNet, thus showing that it successfully improved the estimation performance of DRCFR by effectively correcting the IPW weights by differentiable Pareto smoothing.

Regarding the feature representation learning performance (Figure 4 (a) - (c)), both DRCFR and our method outperformed TARNet. As illustrated in Figure 4 (a), however, with DRCFR and our method, the difference among the absolute parameter values in learned encoder $\Gamma(X)$ decreased, as the number of features $d$ increased, indicating that distinguishing the representation of instrumental variables $\Gamma(X)$ from that of the confounders $\Delta(X)$ was difficult for both methods. This difficulty arises partly because binary treatment $A$ can be accurately predicted solely from the representation of the confounders (i.e., $\Delta(X)$) when there are a sufficient number of confounders $X_\Delta$.

A possible solution is to enforce the independence between feature representations using an additional mutual information regularizer, as with Cheng et al. [2022]. Our future work constitutes the evaluation of such a variant of our method for CATE estimation performance under high-dimensional setups.

---

[3]We used their implementation in https://www.dropbox.com/sh/vrux2exqwc9uh7k/AAAR4tlJLScPlkmPruvbrTJQa?dl=0.

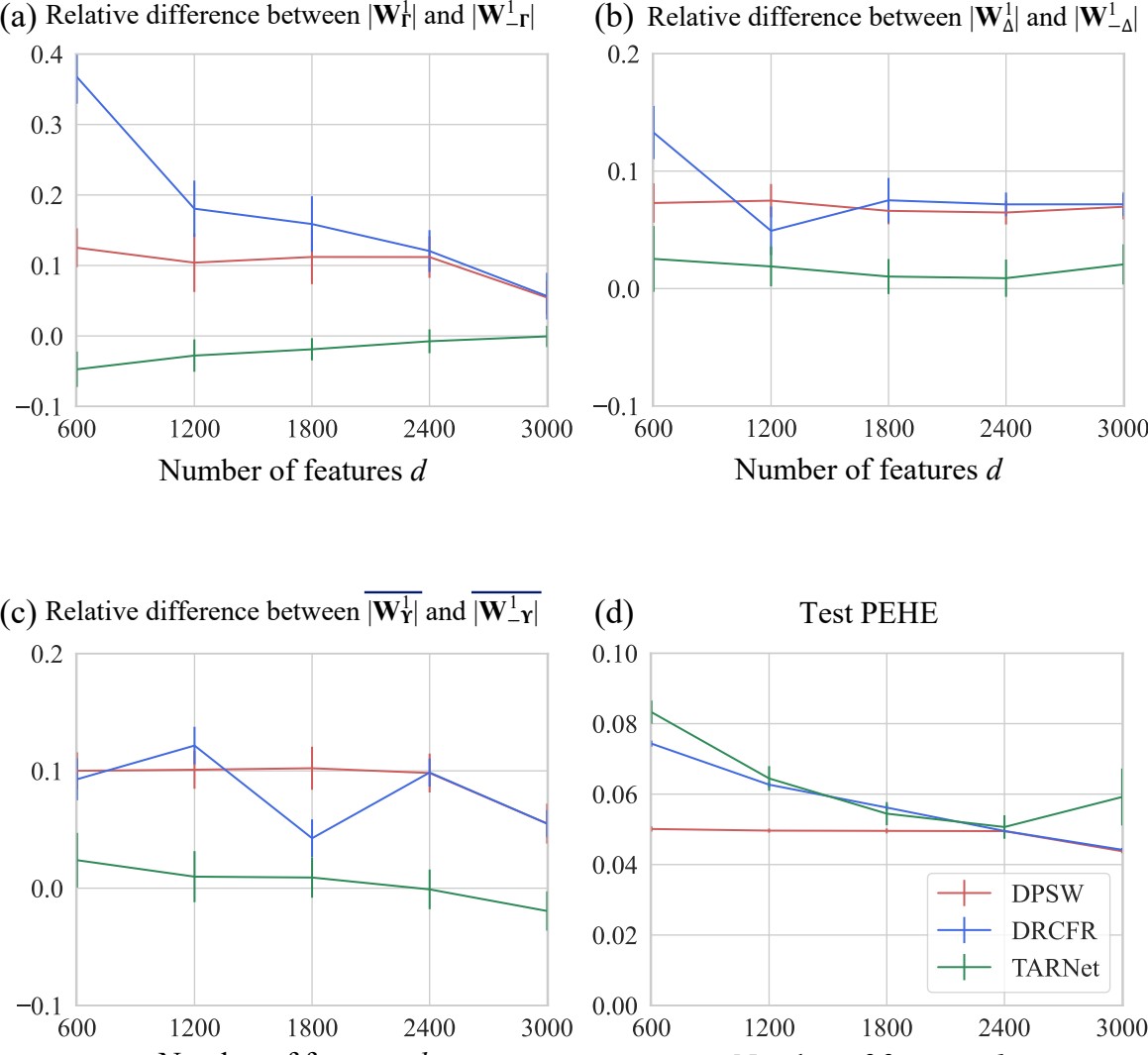

Figure 4: Learned encoder parameter differences and test PEHEs on synthetic data: (a): value difference of $\mathbf{W}^1$ in encoder $\Gamma(\boldsymbol{X})$; (b): value difference of $\mathbf{W}^1$ in encoder $\Delta(\boldsymbol{X})$; (c): value difference of $\mathbf{W}^1$ in encoder $\Upsilon(\boldsymbol{X})$; (d) test PEHEs. With TARNet, since it learns single encoder, we computed all parameter value differences with weight matrix in same encoder.

