# OpenReview forum: "Differentiable Pareto-Smoothed Weighting for High-Dimensional Heterogeneous Treatment Effect Estimation"
_auai.org/UAI/2024/Conference — UAI 2024 poster_

### Official Review · Reviewer_yzhC · 2024-03-23

**Q2-1 Originality-Novelty:** 3
**Q2-2 Correctness-Technical Quality:** 3
**Q2-5 Clarity Of Writing:** 3

**Q1 Summary And Contributions:**

The paper addresses the challenge of estimating treatment effects across individuals using high-dimensional observational data, a problem known as high-dimensional heterogeneous treatment effect estimation. Sample selection bias induced by confounders complicates this task. The authors propose a solution utilizing representation learning to address this bias but highlight a drawback where adjustment variables may be inadvertently eliminated. They introduce a method called disentangled representations for counterfactual regression (DRCFR) to address this issue. However, DRCFR faces challenges with extreme weight values, which can lead to errors in treatment effect estimation. To mitigate this, the authors propose a weight correction framework based on Pareto smoothing, making the estimator numerically stable. They further develop a differentiable Pareto-smoothed weighting framework to replace extreme weight values in a differentiable manner, allowing for end-to-end training.

**Q2-3 Extent To Which Claims Are Supported By Evidence:**

3: Good: the main claims are supported by convincing evidence (in the form of adequate experimental evaluation, proofs, (pseudo-)code, references, assumptions).

**Q2-4 Reproducibility:**

3: Good: key resources (e.g. proofs, code, data) are available and key details (e.g. proofs, experimental setup) are sufficiently well-described for competent researchers to confidently reproduce the main results.

**Q3 Main Strengths:**

- The paper introduces a novel approach to address the challenge of estimating heterogeneous treatment effects in high-dimensional observational data, which is an important and complex problem in fields such as healthcare and advertising.

-  The proposal of using Pareto smoothing to handle extreme weight values contributes to the numerical stability of the estimator, potentially improving the accuracy of treatment effect estimation.

- The development of a differentiable weight correction framework enables end-to-end training of the model, allowing for efficient optimization and potentially better performance.

**Q4 Main Weakness:**

The main concerns is that the paper claims that it is designed to estimate CATE for high-dimensional data. However, it is unclear to me which component in their proposed method addresses this issue.

**Q5 Detailed Comments To The Authors:**

Please clarify more on why the method works well for high-dimensional data in Section 3.

**Q9 Complying With Reviewing Instructions:**

Yes

---

> ### Author Rebuttal · Authors · 2024-04-04
>
> We appreciate the reviewer’s careful read and thoughtful comments. In particular, the question posed by the reviewer is very important and helpful for improving the clarity of our paper. Also, we are glad to hear that the reviewer thinks our proposed framework to be novel and is helpful for an important but complex problem.
>
> Below we provide our answer to the reviewer’s question.
>
> # Important component in our method for high-dimensional setup
>
> > Why the proposed method works well on high-dimensional data?
>
> This is **because it reduces the numerical weight estimation errors by performing Pareto smoothing**.
>
> As described in Section 2.2., the weakness of the DRCFR method is that it suffers from huge weight estimation error due to the computation of the inverse of propensity score. This issue can occur even in a low-dimensional setup; however, it is more serious in high-dimensional cases, due to the difficulty of correctly estimating the propensity score model from high-dimensional features [1]. Hence, it is crucial to improve the numerical robustness by decreasing the weight estimation error.
>
> Pareto smoothing is a suitable technique for this purpose because it can effectively correct extreme weight values and construct a numerically robust estimator. However, directly applying this technique is impossible due to its non-differentiable nature. For this reason, we have proposed a differentiable Pareto-weighting framework that allows for end-to-end training.
>
> We thank the reviewer again for pointing out this important point. **We will revise the description in Section 3 based on the above discussion**.
>
> > [1] Assaad, Serge and Zeng, Shuxi and Tao, Chenyang and Datta, Shounak and Mehta, Nikhil and Henao, Ricardo and Li, Fan and Carin, Lawrence. Counterfactual representation learning with balancing weights. AISTATS, 2021.

---

### Official Review · Reviewer_4wSD · 2024-03-24

**Q2-1 Originality-Novelty:** 3
**Q2-2 Correctness-Technical Quality:** 3
**Q2-5 Clarity Of Writing:** 3

**Q1 Summary And Contributions:**

This paper studies a novel but important problem, i.e., estimation of HTE towards high-dimensional covariates. To this end, authors build upon the framework of DR-CFR, which utilizes IPW to balance covariates across groups. However, the drawback of IPW is the instability of propensity score, especially in the case of high-dimensional covariates. Hence, authors propose to adopt Pareto-smoothed framework to overcome this issue, together with some differential approaches.

**Q2-3 Extent To Which Claims Are Supported By Evidence:**

3: Good: the main claims are supported by convincing evidence (in the form of adequate experimental evaluation, proofs, (pseudo-)code, references, assumptions).

**Q2-4 Reproducibility:**

3: Good: key resources (e.g. proofs, code, data) are available and key details (e.g. proofs, experimental setup) are sufficiently well-described for competent researchers to confidently reproduce the main results.

**Q3 Main Strengths:**

The paper is well-organized with clear presentation.
The problem is practical and novel.
The overall framework is solid.

**Q4 Main Weakness:**

1.	According to previous results, adopting adjustment variables, i.e., predictive variables, is beneficial for achieving efficient estimators [1], why this paper aims to separate such variables from confounders?
2.	Why choosing DR-CFR? I would be more pleased to see a more detailed explanation.

[1] Andrea Rotnitzky and Ezequiel Smucler. 2020. Efficient Adjustment Sets for Population Average Causal Treatment Effect Estimation in Graphical Models. J. Mach. Learn. Res. 21, 188 (2020), 1–86.’

**Q5 Detailed Comments To The Authors:**

See in Weakness.

**Q9 Complying With Reviewing Instructions:**

Yes

---

> ### Author Rebuttal · Authors · 2024-04-04
>
> We cordially thank the reviewer for posing the two questions about our proposed weighting framework, both of which are useful to further enhance the clarity of our paper. We are very glad to see that the reviewer has found our proposed framework novel, important, solid, and practical.
>
> Below we respond to the reviewer’s questions.
>
> # The reason why we perform feature separation
>
> > Q1. According to previous results, adopting adjustment variables, i.e., predictive variables, is beneficial for achieving efficient estimators [1], why this paper aims to separate such variables from confounders?
>
> Let us clear up possible confusions. Our goal is **NOT** to remove adjustment variables, but to **adopt them while removing the sample selection bias** due to confounders, as with the reviewer’s cited literature [1].
>
> This goal, however, is difficult to achieve because **in our problem setup, it is unclear which variable in (high-dimensional) input features $X$ corresponds to a confounder or an adjustment variable**. Since adjustment variables are not related to sample selection bias, naively converting all features $X$ into a feature representation that is balanced between treated and untreated individuals will unnecessarily lose helpful information for potential outcome prediction.
>
> For this reason, **we perform data-driven feature separation via weighted representation learning to obtain the neural-network-based feature representation of confounders and that of adjustment variables from input features $X$**. By removing the sample selection bias based on the former and adopting the latter directly for potential outcome prediction, our method achieves good CATE estimation performance.
>
> # The choice of weighted representation learning methods
>
> > Q2. Why choosing DR-CFR?
>
> The main reason is described in Related Work section (Section 5): **The DR-CFR method focuses on CATE estimation and is founded on weighted representation learning, which is a promising approach for high-dimensional setup**.
>
> Indeed, there are only a few data-driven feature separation methods for treatment effect estimation, and most of them are designed for average treatment effect (ATE), not CATE. An exception is some recent work [a], which aims to improve the DR-CFR by enforcing the independence of neural-network-based feature representations. Incorporating such a recent idea into the proposed weighting framework is left as our future work.
>
> > [a] Cheng, M., Liao, X., Liu, Q., Ma, B., Xu, J., & Zheng, B. Learning disentangled representations for counterfactual regression via mutual information minimization. In Proceedings of the 45th International ACM SIGIR Conference on Research and Development in Information Retrieval (pp. 1802-1806), 2022.

---

### Official Review · Reviewer_GGDW · 2024-03-25

**Q2-1 Originality-Novelty:** 2
**Q2-2 Correctness-Technical Quality:** 3
**Q2-5 Clarity Of Writing:** 3

**Q1 Summary And Contributions:**

The paper proposes weight smoothing method for end to end training to estimate CATE. The paper demonstrates the performance with both semi-synthetic and synthetic data.

**Q2-3 Extent To Which Claims Are Supported By Evidence:**

3: Good: the main claims are supported by convincing evidence (in the form of adequate experimental evaluation, proofs, (pseudo-)code, references, assumptions).

**Q2-4 Reproducibility:**

3: Good: key resources (e.g. proofs, code, data) are available and key details (e.g. proofs, experimental setup) are sufficiently well-described for competent researchers to confidently reproduce the main results.

**Q3 Main Strengths:**

1. The paper is extremely well written in my opinion with clear problem setup, motivation, techniques used and experimental details.
2. The method proposed is an end-to-end training pipeline, which is easily scalable.
3. Overall, the paper solves an important problem.

**Q4 Main Weakness:**

1. I like reading the paper and the idea behind the algorithm. However, it occurs to me that only the idea of applying Pareto smoothing to a specific method: DRCFR is new. So even though the paper is well written, I think this deminishes the originality of the paper.
2. As the main contribution of the paper is the smoothing scheme, I think the paper should also compare with other methods dealing with extreme weights, e.g. the truncation you mentioned or just throwing away units with extreme weights (see Doubly-robust and heteroscedasticity-aware sample trimming for causal inference by Khan and Ugander).

**Q5 Detailed Comments To The Authors:**

Major:
1. What is the heuristic used to select $M$?
2. In your experiments, which hyperparameter is the modt important? I think more insights would be helpful for people using the method.
3. For other meta-learner baselines, did you consider other base learners?
4. Could you provide some intuitions on why self-normalization helps?
5. The SD of the results seem large, maybe consider increase the number of replications.

Minor:
1. In page 1, 'Moreover, due to the lack of prior knowledge about features, it is difficult for practitioners to correctly separate adjustment variables from confounders.' I think this is not true in general. For example in medical applications, practitioners have a lot of domain knowledge, the main issue might be non-compliance.
2. Typo on page 3: 'To achieve high CATE estimation performance, it is essential how to compute the weight value,'

**Q9 Complying With Reviewing Instructions:**

Yes

---

> ### Author Rebuttal · Authors · 2024-04-04
>
> We appreciate many constructive suggestions by the reviewer, most of which are very helpful for improving the reliability of experimental evaluation. Below we present several additional experimental results.
>
> # Comparison with other weighting schemes
>
> Let us point out that **we have already evaluated the performance with truncated weights** (**DRCFR Trunc.** in Table 1). Regarding another weighting scheme (i.e., ignoring units with extreme weights), we tested it using the same threshold with DRCFR Trunc. The following table shows that our DPSW achieved lower test PEHE than this baseline, demonstrating that **by smoothening the extreme weights, our method can effectively use the information of weighted loss** for potential outcome prediction.
>
> || News | ACIC |
> | :--- | ---- | ---- |
> | DPSW | 2.20 $\pm$ 0.72 | 0.57 $\pm$ 0.03 |
> | DRCFR Trunc. | 2.42 $\pm$ 0.79 | 1.06 $\pm$ 0.06 |
> | DRCFR Ignore | 2.35 $\pm$ 0.75 | 0.84 $\pm$ 0.06 |
>
>
> # Number of replaced weights $M$ (Q. 1)
>
> As with the original Pareto smoothing [1], we set $M$ using sample size (See Eq. (8)). In experiments, we used mini-batch sample size for this purpose.
>
> > [1] Aki Vehtari, Daniel Simpson, Andrew Gelman, Yuling Yao, and Jonah Gabry. Pareto smoothed importance sampling. Journal of Machine Learning Research (JMLR; accepted for publication), 2024.
>
>
> # Important hyperparameter (Q. 2)
>
> The most important hyperparameter is **$\varepsilon$ in differentiable rank function $r_{\varepsilon}$**. It controls the trade-off between approximation precision and smoothness: As shown in Figure 2, $\varepsilon \rightarrow 0$ makes the rank function non-differentiable but offers the exact rank values. As described in Section 4.1, we tuned this hyperparameter based on the objective function value on validation data; such tuning is standard for CATE estimation.
>
> # Further comparison with meta-learner baselines (Q. 3)
>
> We considered random forest (RF), gradient boosting (GB), and support vector machine (SVM) as the base learners. Table 1 only shows the results with RF because we empirically observed its good performance. We confirmed this by performing additional experiments.
>
> The following table shows that RF was the best, though some results were very similar to those with other base learner. However, **none of them were better than our DPSW method**.
>
> || News | ACIC |
> | :--- | ---- | ---- |
> | SL (RF) | 2.83 $\pm$ 1.11 | 1.69 $\pm$ 0.52 |
> | SL (GB) | 2.90 $\pm$ 1.05 | 1.86 $\pm$ 1.12 |
> | SL (SVM) | 4.31 $\pm$ 1.26 | 2.23 $\pm$ 0.40 |
> | TL (RF) | 2.55 $\pm$ 0.82 | 2.23 $\pm$ 0.50 |
> | TL (GB) | 2.68 $\pm$ 0.88 | 2.29 $\pm$ 0.47 |
> | TL (SVM) | 3.33 $\pm$ 1.27 | 3.15 $\pm$ 0.10 |
> | XL (RF) | 2.77 $\pm$ 1.01 | 1.05 $\pm$ 0.72 |
> | XL (GB) | 2.99 $\pm$ 1.11 | 2.49 $\pm$ 0.18 |
> | XL (SVM) | 3.33 $\pm$ 1.27 | 2.29 $\pm$ 0.07 |
> | DRL (RF) | 23.9 $\pm$ 5.96 | 3.77 $\pm$ 8.96 |
> | DRL (GB) | 27.2 $\pm$ 6.97 | 12.1 $\pm$ 5.03 |
> | DRL (SVM) | 30.7 $\pm$ 9.91 | 3.85 $\pm$ 2.11 |
> | DPSW | 2.20 $\pm$ 0.72 | 0.57 $\pm$ 0.03 |
>
> # Intuition about self-normalized weights (Q. 4)
>
> By dividing each weight’s value by its empirical mean, **self-normalization prevents the weights from being too small or too large relative to each other**, thus reducing the variability in the estimator and hence leading to lower estimation variance.
>
> # Standard deviation (SD) (Q. 5)
>
> As the reviewer pointed out, the SD of the results is relatively large, particularly on News dataset. This is not due to the number of data replications because we have already used all the available simulated data for News dataset.
>
> A possible reason is that **unlike the existing work (e.g., [2]), we performed no dimension reduction of features X**, such as principal component analysis (PCA). Performing PCA beforehand would greatly facilitate the estimation problem because the outcomes of News data were simulated only from the low-dimensional features given by the pre-trained topic model (since the number of topics is small, this model roughly corresponds to PCA).
>
> However, since outcome is not always determined by such low-dimensional features, performing dimension reduction beforehand will lose important feature information for CATE estimation, if one cannot select a suitable dimension reduction method. For this reason, we have directly inputted all features X without performing any dimension reduction methods, thus tackling a challenging high-dimensional setup.
>
> > [2] Johansson, Fredrik, Uri Shalit, and David Sontag. Learning representations for counterfactual inference. ICML, 2016.
>
> # More practical real-world scenarios (Minor 1.)
>
> Once again, we thank the reviewer for a very constructive comment!
>
> **It is true that the prior knowledge about features is less available in non-compliance setting** because even in medical treatment scenarios, it is often unclear why each individual does not follow the treatment assignment. We appreciate the reviewer’s comment and would like to consider such a setting in our future work.

---

### Official Review · Reviewer_zCMu · 2024-03-25

**Q2-1 Originality-Novelty:** 3
**Q2-2 Correctness-Technical Quality:** 3
**Q2-5 Clarity Of Writing:** 4

**Q1 Summary And Contributions:**

This paper proposes an approach for high-dimensional heterogeneous treatment effect estimation. In particular, based on the DRCFR method, they use Pareto smoothing that replaces IPW extreme weights with more stable weights. Then, they use a differentiable approximation technique for computational efficiency. They show that the proposed methods beat several benchmarks on both semi-synthetic and synthetic datasets.

**Q2-3 Extent To Which Claims Are Supported By Evidence:**

3: Good: the main claims are supported by convincing evidence (in the form of adequate experimental evaluation, proofs, (pseudo-)code, references, assumptions).

**Q2-4 Reproducibility:**

3: Good: key resources (e.g. proofs, code, data) are available and key details (e.g. proofs, experimental setup) are sufficiently well-described for competent researchers to confidently reproduce the main results.

**Q3 Main Strengths:**

I think the paper is generally well-written. I think the paper motivates the approach and explains the components of their approach well. It’s very readable and understandable for non-experts and I think their method is very intuitive. Also, the presentation of their experimental results is very clear.

**Q4 Main Weakness:**

The main weakness of this paper is its technical contribution. In particular, it seems that this paper is combining several different methods, but no proof of consistency of their estimates or inference is provided. It only shows experimental results although empirically it shows a gain of their method over benchmarks.

**Q5 Detailed Comments To The Authors:**

- What is a disadvantage of using GPD other than computing the rank of IPW weights? Even with this, it still seems that GPD stabilizes the extreme weights of IPW by losing nothing.
- The approach provided by the authors combines several techniques from several papers, e.g. DRCFR, Pareto smoothing, and differentiable ranking techniques. Could the authors describe the assumptions for these methods and why combining all of them will not violate any assumptions?

**Q9 Complying With Reviewing Instructions:**

Yes

---

> ### Author Rebuttal · Authors · 2024-04-04
>
> We are deeply grateful to the reviewer for their insightful comments, which provided us with a great opportunity to reconsider the theoretical aspects of our work. We are very glad to hear that the reviewer thinks our paper very readable, very understandable, well-motivated and very intuitive.
>
> We understand the reviewer’s concerns about the lack of theoretical results in our paper. Unfortunately, to the best of our knowledge, **the literature on (weighted) representation learning methods for CATE estimation usually does not offer the asymptotic consistency results**, due to the lack of learning theory on complex neural network models.
>
> To resolve the reviewer’s concerns as much as possible, below we explain several theoretical results on Pareto-smoothed weighting based on the original article [1].
>
> > [1] Aki Vehtari, Daniel Simpson, Andrew Gelman, Yuling Yao, and Jonah Gabry. Pareto smoothed importance sampling. Journal of Machine Learning Research (JMLR; accepted for publication), 2024.
>
> # (A) Asymptotic consistency
>
> >this paper is combining several different methods, but no proof of consistency of their estimates or inference is provided.
>
> Regarding Pareto smoothing, the original article [1] provides the asymptotic consistency results: The empirical average weighted by Pareto-smoothed importance sampling weights is guaranteed to converge to the true expected value under some mild conditions (Theorem 1 in [1]).
>
> However, when we employ these Pareto-smoothed importance sampling weights in the weighted representation learning, **it is extremely difficult to prove the asymptotic convergence to the true CATE value, due to the complexity of neural network models and the lack of the theoretical understanding about their learning theory**. For this reason, it is quite usual that the weighted representation learning methods, including DRCFR [2] and our method, rely on the empirical performance comparison.
>
> Note that the differentiable ranking technique itself does not involve the asymptotic consistency: It offers an approximation of (non-differentiable) rank function, whose approximation precision is determined not by sample size but by hyperparameter $\varepsilon$, which we tuned with validation data (See Figure 2 for its illustration).
>
> > [2] Hassanpour, Negar, and Russell Greiner. "Learning disentangled representations for counterfactual regression." ICLR. 2020.
>
> # (B) Possibility of assumption violation
>
> The main assumptions of each component can be summarized as follows.
>
> 1. **Pareto smoothing**: The asymptotic consistency results in [1] require the condition that **the distribution of importance sampling weight is absolutely continuous**. In our setup, this condition requires the distribution of IPW weight to be absolutely continuous.
> 2. **DRCFR**: This method needs the **graphical model assumption** (shown in Figure 1) that **for instance, treatment $A$ is determined by functions $\Gamma(X)$ and $\Delta(X)$**, both of which are formulated as neural network encoders.
>
> **Combining these assumptions leads to a condition that weight $w_i \propto \frac{1}{\pi_{a_i}(\Gamma(x_i), \Delta(x_i))}$ in Eq. (4) follows an absolutely continuous distribution**. Under binary treatment ($A = 0$ or $A=1$), the weight distribution is a mixture of the two distributions of IPW weights; in each distribution, the weight is given by propensity score model $\pi_{0}(\Gamma(X), \Delta(X))$ or $\pi_{1}(\Gamma(X), \Delta(X))$, respectively.
>
> **In many cases, either of Assumptions 1 and 2. will not violate the other**. An exception would be, for instance, the cases where we formulate the neural network model for propensity score $\pi$ to be discontinuous with respect to its inputs.
>
> Regarding differentiable ranking technique, it imposes no assumption, but we need to tune its hyperparameter (with validation data) to strike a good balance between the approximation precision and the smoothness (See Figure 2 for the illustration of this balance).
>
>
> # (C) Disadvantage of using generalized Pareto distribution (GPD)
>
> A possible disadvantage is that **it is not trivial to well fit the GPD parameters to the weight values**, implying that the weight smoothing may fail due to the bad model fit. As a solution to this issue, the authors of Pareto smoothing [1] have empirically demonstrated that we can effectively evaluate the reliability of GPD fitting by employing the estimated value of GPD’s shape parameter $\xi$, which determines the heaviness of the distribution tail. Using such a diagnostic, for instance, they conclude that the Pareto-smoothing-based estimate is likely to be invalid if $\hat{\xi} > 1$ (See Section 6 in [1] for details).
>
> **However, by fitting the GPD parameters with the standard moment-matching method (called the probability weighted moment (PWM)), we empirically observed good CATE estimation performance**. Our future work will investigate the application of more efficient GPD parameter estimation methods.

---

### Official Review · Reviewer_nneT · 2024-03-28

**Q2-1 Originality-Novelty:** 3
**Q2-2 Correctness-Technical Quality:** 3
**Q2-5 Clarity Of Writing:** 4

**Q1 Summary And Contributions:**

The work estimates heterogeneous treatment of individuals from high-dimensional data. The propose the use of a previously proposed Pareto-weighted framework in their pipeline. However, their the objectives to optimize being non-differentiable, they find a workaround of  using differentiable ranking. Finally they use a data-driven feature separation method for their effect estimation.

**Q2-3 Extent To Which Claims Are Supported By Evidence:**

3: Good: the main claims are supported by convincing evidence (in the form of adequate experimental evaluation, proofs, (pseudo-)code, references, assumptions).

**Q2-4 Reproducibility:**

2: Fair: key resources (e.g. proofs, code, data) are unavailable but key details (e.g. proof sketches, experimental setup) are sufficiently well-described for an expert to confidently reproduce the main results.

**Q3 Main Strengths:**

Disclaimer: This line of work is completely new to me. Might have been the term "Pareto" for the paper getting referred to me.

Having said that, it was a very nice read and understandable. All the methods used as inspiration for their pipeline or for comparison are well noted out. If they make any assumptions or choices, those are justified as well.

**Q4 Main Weakness:**

N/A

**Q5 Detailed Comments To The Authors:**

Could you please provide a runtime table of methods? Complexity analysis of all the other methods might not be possible, but can be done for the proposed method with an additional $O(num\_iterations\_till\_convergence)$ factor.

**Q9 Complying With Reviewing Instructions:**

Yes

---

> ### Author Rebuttal · Authors · 2024-04-04
>
> We thank the reviewer for the suggestions about computation time comparison, which are very helpful to improve the quality of our work. We are very pleased to see that the reviewer has found our paper to be a very nice read and its clarity to be excellent.
>
> To respond to the request by the reviewer, below we present the runtime comparison results and compare the time complexity of the original weighted representation learning method, DRCFR, and our proposed Pareto-weighted framework.
>
> # Runtime comparison
>
> We performed training run time comparison, using a single sample in the News dataset. Here we employed a 64-bit UbuntuOS machine with 2.90GHz Intel(R)Gold 622R (x2) CPUs and 1024-GB RAM.
>
> || News [sec.] |
> | :--- | ---- |
> | LR-1 | 1.29 |
> | LR-2 | 3.07 |
> | SL | 86.61 |
> | TL | 62.03 |
> | XL | 140.81 |
> | DRL | 741.17 |
> | CF | 13.79 |
> | CF DML | 83.65 |
> | TARNet | 11.04 |
> | GANITE | 29.00 |
> | DRCFR | 74.69 |
> | DRCFR Norm. | 76.64 |
> | DRCFR Trunc. | 76.08 |
> | DRCFR Ignore | 71.08 |
> | PSW | 74.55 |
> | DPSW | 89.68 |
> | DPSW Norm. | 88.91 |
>
> (Following the suggestion by Reviewer GGDW, here we added an additional baseline, DRCFR Ignore, which simply forces the extreme weight values to be zero.)
>
> Note that the run time hugely depends on the implementation. For instance, TARNet is implemented with Python library Jax, while our DPSW methods use PyTorch library, making the exact comparison difficult.
>
> # Complexity analysis
>
> The above run time table illustrates that our DPSW needed a bit longer computation time than DRCFR.
>
> The computational difference between these methods is whether to perform Pareto-smoothed weighting. That is, **our method requires additional time of computing Pareto-smoothed weights to evaluate the objective function and its gradient for each iteration in training phase**. **This time complexity is $\mathcal{O}(B log B)$ for mini-batch size $B$** (needed to compute differentiable rank function, as described in Section 3.4.1).

---

### Meta-Review · Area_Chair_Rxgr · 2024-04-12

This work presents a differentiable Pareto-smoothed weighting framework for CATE estimation. Though supporting theory for the proposed approach was not provided, reviewers stated that the proposed approach was clearly presented and tackles an important problem. There was good engagement between authors and reviewers during the rebuttal period.

Here is a more detailed summary on reviewers' opinions on this work. They noted the following main strengths:
- Well written: nneT, zCMu, GGDW, 4wSD
- Clearly presented: nneT, zCMu, GGDW, 4wSD
- Solves an important/practical/novel problem: GGDW, 4wSD, yzhC
- Easily scalable / efficient: GGDW, yzhC
- Numerically stable: yzhC

Reviewers noted the following main weaknesses:
- technical contribution / lack of proof of consistency: zCMu (partially addressed in response)
- limited novelty: GGDW (not addressed in response)